# Selection rules for the S-Matrix bootstrap

Anjishnu Bose,* Aninda Sinha† and Shaswat S. Tiwari‡

*Centre for High Energy Physics, Indian Institute of Science,*

*C.V. Raman Avenue, Bangalore 560012, India.*

March 11, 2021

### Abstract

We examine the space of allowed S-matrices on the Adler zeros' plane using the recently resurrected (numerical) S-matrix bootstrap program for pion scattering. Two physical quantities, an averaged total scattering cross-section, and an averaged entanglement power for the boundary S-matrices, are studied. Emerging linearity in the leading Regge trajectory is correlated with a reduction in both these quantities. We identify two potentially viable regions where the S-matrices give decent agreement with low energy S- and P-wave scattering lengths and have leading Regge trajectory compatible with experiments. We also study the line of minimum averaged total cross section in the Adler zeros' plane. The Lovelace-Shapiro model, which was a precursor to modern string theory, is given by a straight line in the Adler zeros' plane and, quite remarkably, we find that this line intersects the space of allowed S-matrices near both these regions.

# Contents

---

*anjishnubose98@gmail.com

†asinha@iisc.ac.in

‡shaswat10041998@gmail.com

# 1  Introduction

Before the advent of QCD, Chew's S-matrix bootstrap program [1] was at the forefront of research in the 1960s. One of the most studied questions was finding a bootstrap solution to pion scattering, which was consistent with Lorentz invariance, crossing symmetry and which could produce the phenomenologically observed Regge trajectories for the light mesons (Chew-Frautschi plot). String theory originated in an attempt to find such a solution, leading to the Veneziano amplitude [2] and its generalizations, particularly the so-called Lovelace-Shapiro model [3] for pion scattering. Unitarity led to the Regge intercept of unity, while phenomenology demanded that the Regge intercept be near half. String theory ideas took off in a different direction, leading to the identification of consistent string theories as quantum theories of gravity,–see [4] for a very nice account of the early history of string theory. This original attempt to connect with hadron physics was more or less abandoned until the discovery of the AdS/CFT correspondence,–see [5] for a recent review of holography inspired string hadron physics. The question of whether the bootstrap could give a consistent picture of hadron physics thus lay unanswered until the current re-examination of this question through the papers [6, 7].

In this work, we will closely follow the numerical methods initiated in [6, 7] to study the space of allowed S-matrices, allowing for some interesting modifications. The ingredients we will borrow from [6] are a) using a crossing symmetric basis which took into account the cut at $s = 4$ in the complex s-plane b) imposing Adler zeros in the isospin-0 and isospin-2, spin-0 partial waves, and crucially c) Imposing the $\rho$ resonance at $\sqrt{s} = (5.5 - 0.5i)$ in units where $m_\pi = 1$. Using these ingredients and demanding partial wave unitarity, an exclusion region called the "lake" was found in the space of Adler zeros. In [8], we supplemented these conditions by imposing the signs on the D-wave scattering lengths dictated by unitarity and the Froissart-Gribov formula. In addition, we set the same signs on the linear combinations of S-wave scattering lengths, which follow from chiral perturbation theory ($\chi PT$). Equivalently, these signs follow from demanding certain sign-definiteness in the quantum part of relative entropy, as explained in [8]. A narrower allowed region called the "river",–see fig.(1)–was obtained.

With such a huge class of potentially interesting S-matrices, a natural question is which of these

boundary S-matrices exhibit linear Regge trajectories (we will refer to this as linearity frequently) and are compatible with the experimental S- and P-wave scattering lengths [9]. We will focus on the leading Regge trajectory. Quite fascinatingly, the regions along the river bank which admit linearity are quite limited. In particular, one region is close to the Adler zero values, which follow from two loop $\chi PT$. Another small region with linearity and decent S- and P-wave scattering lengths lies in the lower boundary, far from the $\chi PT$ values. More remarkably, both these regions also coincide with where the Lovelace-Shapiro model passes through the allowed space of S-matrices. In the Lovelace-Shapiro (LS) model, the slope and intercept can be adjusted to allow Adler zeros in the isospin-0 and isospin-2, spin-0 partial waves. This gives a line of models that intersects the river in distinct places. The zero-width LS model itself is *not* unitary in these interesting regions[1]. The bootstrap approach potentially leads to a unitary, finite width version of the LS model.

Once these observations are in place, one of the main questions arises: What is so unique about the standard model, or less ambitiously, the models exhibiting linearity? While the two-loop $\chi PT$ point lies within the river and hence, in the current formulation of the bootstrap, is challenging to study directly[2], we can ask what is so special about the kink type feature near this point. We find that if we consider the averaged total scattering cross-section, $\bar{\sigma}$, which is related to the imaginary part of the $AB \to AB$ type amplitude in the forward limit via the optical theorem, for individual boundary S-matrices, then this standard model kink is the region where the sharpest decrease in $\bar{\sigma}$ happens. A related observation is that the isospin space entanglement entropy (more appropriately the entanglement power to be described below) for the final state particles in the forward direction also exhibits a similar reduction. These observations hint at a natural quantum information-theoretic selection principle in the space of allowed S-matrices.

## 2 S-matrix bootstrap reloaded

Let us begin by briefly recalling the key numerical ideas used in [6]. For more details, we refer the reader to Appendix A. We are interested in pion-pion scattering in $3 + 1$ dimensions, where the S-matrix is decomposed into the isospin channels. For numerical purposes, using the technology developed in [7], a crossing symmetric basis is used, which encapsulates the $s$-channel cut at $s = 4$. A corresponding partial wave expansion is done, and partial wave unitarity is checked. The low energy Adler zeros are imposed on the isospin-0 and isospin-2, spin-0 partial wave coefficients. These are at unphysical values of the Mandelstam variable $s$ and are treated as parameters to vary. State of the art two-loop $\chi PT$ [11] places these zeros at $s_0 = 0.4195, s_2 = 2.008$ which provides a comparison point. We will sometimes refer to this as the "standard model point" and the kink in the neighbourhood of this, as the "standard model kink". Note that the former is an abuse of terminology since the location of the standard model Adler zeros, non-perturbatively, is of course not known. The $\rho$ resonance at $\sqrt{s} = 5.5 - 0.5i$ is imposed as a zero on the physical sheet. Using these, an exclusion region dubbed as the "lake" was obtained in [6]. On further imposing the experimental S and P-wave scattering lengths as inputs, a smaller region dubbed the "peninsula"

---

[1] See for example, [10].

[2] Since each point inside the river corresponds to infinitely many allowed S-Matrices, there is no unique S-Matrix which describes two-loop $\chi PT$ in our current formulation.

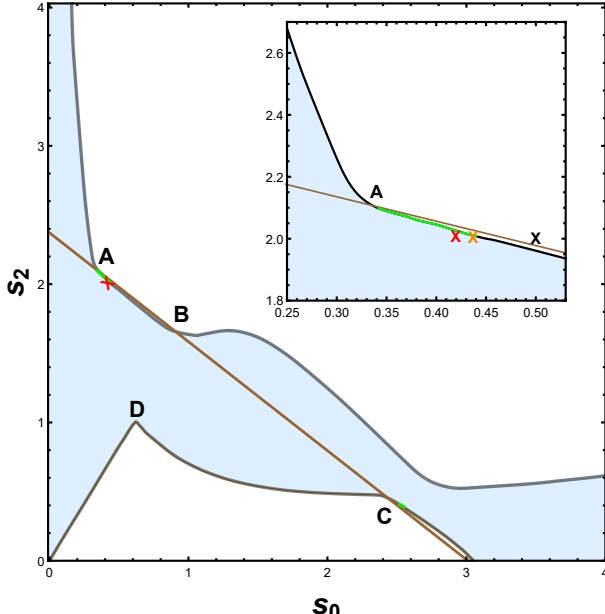

Figure 1: Pion river at $N_{max} = 16$. The behaviour changes rapidly at points A,B,C and D. The red cross marks the two-loop $\chi PT$. The brown straight line is the Lovelace-Shapiro model allowing for general Adler zeros. The green regions marks exhibit linearity. The inset shows a zoomed version with the tree-level, one-loop and two-loop $\chi PT$ values indicated in black, orange and red respectively.

was also obtained. Borrowing a terminology from the numerical conformal bootstrap, a "kink" was identified at $(s_0, s_2) \approx (0.36, 2.04)$. Following [6], we will truncate the crossing symmetric basis at $N_{max}$ and impose unitarity on $L_{max}$ partial waves on an $s$-grid of 200 points.

In [8], in addition to the conditions above, dispersion relations and $\chi$PT motivated inequalities constrained the lake further leading to a "river" like allowed region. The D-wave inequalities were found using the Froissart-Gribov representation which rely on subtractions–(for details refer to [12]),– and are of the form,

$$a_2^{(0)} + 2a_2^{(2)} \geq 0, \quad a_2^{(0)} - a_2^{(2)} \geq 0. \tag{2.1}$$

The S-wave inequalities were motivated from 2-loop $\chi PT$ and are:

$$a_0^{(0)} + 2a_0^{(2)} \geq 0, \quad a_0^{(0)} - a_0^{(2)} \geq 0, \quad a_0^{(2)} \leq 0. \tag{2.2}$$

The first two S-wave inequalities are the analogs of the D-wave inequalities but do not have a dispersion relation proof[3]. In addition to these inequalities, $\chi$PT results also suggest $a_2^{(2)} \geq 0$. The S-wave inequalities lead to stronger constraints than the D-wave ones; this is because of the form of the ansatz used in the analysis in [6]. In section 5, we will discuss S-matrix bootstrap without imposing the S-wave or D-wave inequalities.

The river is indicated in fig.(1). We have indicated by **A, B, C, D**, four obviously interesting points where there is some sharp change in behaviour, as will be elaborated below. The point **A**

---

[3]Zvi Bern tells us that if there is a dimensional continuation of the dispersion relations that enables us to bypass subtractions, then one should be able to impose the inequalities that follow from unsubtracted dispersion relations. We will leave a detailed investigation of this possibility for future.

is what we will refer to as the standard model kink and is given by $(s_0, s_2) = (0.33, 2.12)$[4]. This is different from the "kink" in [6] since, barring the signs on the scattering lengths explained above, we have not incorporated anything from experiments. Note that the two-loop $\chi$PT value for the zeros is quite close to the kink. It appears that the tree-level, one-loop, and two-loop zeros are slowly moving towards the kink **A**.

The straight brown line in fig.(1) is the zero-width (non-unitary) Lovelace-Shapiro (LS) model extended to allow for Adler zeros as described in detail in Appendix C. The LS model equation in the plane of Adler zeros is given by $s_2 \approx 2.37 - 0.79\, s_0$ and intersects the river at several locations as can be seen in fig.(1). These intersection points cause a remarkable change in the behaviour of the river boundary. The two intersections (**A** and **B**) are on the upper boundary, and the third intersection (**C**) is on the lower boundary. Apart from the intersection, there is also a tip like structure on the lower boundary at **D**.

Next, we will check for the resonances along the river.

# 3 Linear Regge trajectory

We shall use the methods developed in Appendix B.1 to determine the location of resonances in partial waves. As can be seen in eq.(B.8), resonances correspond to peaks in $|f_\ell|^2$. Since we do not have elastic unitarity, we will track peaks in the ratio $|f_\ell|^2/|S_\ell|^2$. Curiously enough, there is a small region around the kink near **A** (specifically, we have observed almost linear Regge for $s_0 \in (0.34, 0.44)$) of fig.(1), where we observe discernible peaks for all spins (we have checked up to $\ell = 8$.). When we plot their locations vs. spin, $\ell$, they are approximately linear (with the statistical coefficient of determination $R^2 \geq 0.9$). Furthermore, they are relatively close to their corresponding experimental values, as shown in fig.(2) and fig.(8). As shown in fig.(2.b) the slopes for the even and odd peaks coincide only near **A**.

In fig.(1), there is also a region near **C**, which has approximate linearity of resonances. However, the area where this linearity holds is small in comparison to **A**, and the individual peak values are somewhat further away from experimental values, as can be seen in fig.(8). Nevertheless, we do not have a definitive way to rule out the S-matrices in this region as unfeasible for describing experiments. In region **B**, curiously, even/odd spins line up separately with different slopes (see Appendix B.2). Finally, in region **D**, there is approximate linearity with a larger slope; however, $\ell = 8$ is missing. The bottom line is that there is interesting linearity of different kinds in all four discernibly interesting regions in fig.(1).

One might wonder whether these peaks are simply numerical artifacts. To put these unsavoury thoughts to rest, we shall show convergence with $N_{max}$ and $L_{max}$ in Appendix D. It is somewhat challenging to demonstrate convergence with $N_{max}$, since the river boundary changes slightly with $N_{max}$. Also, as can be seen in fig.(8), several discontinuous jumps in peak positions can alter the best fit and $R^2$ values significantly. The $L_{max}$ convergence is somewhat easier to demonstrate since the river changes considerably less with $L_{max}$.

---

[4]**B**: (0.92,1.66), **C**:(2.43,0.46), **D**:(0.62,1.0) .

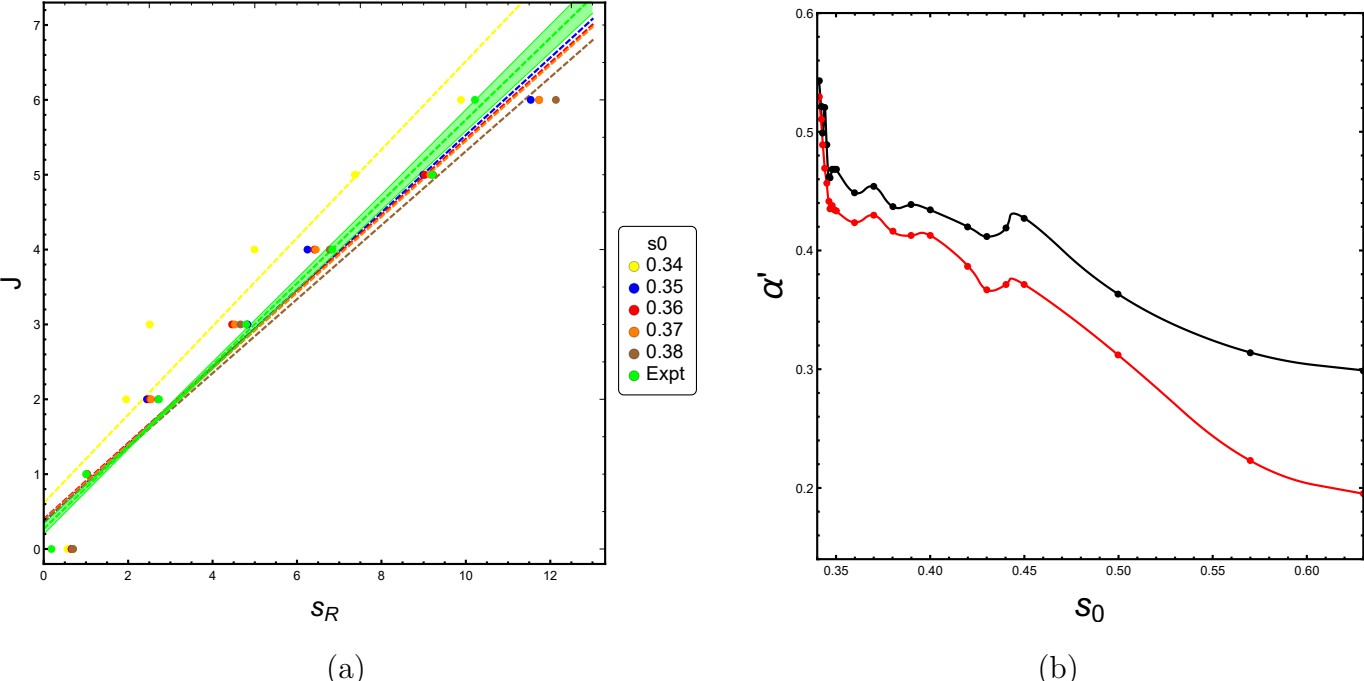

Figure 2: (a) Variation of best fit line with $s_0$ on the upper boundary. For $s_0 = 0.35$ the best fit line including $\ell = 0$ to $\ell = 6$ is given by $J = 0.38 + 0.51s_R$ while the experimental one is $J = 0.27 + 0.54s_R$. (b) Variation of the slope $\alpha'$ with $s_0$ on the upper boundary in the neighbourhood of **A**. Except near $s_0 \approx 0.35$ the even/odd spins separate.

# 4 Selecting the standard model

One of the main lessons that the conformal bootstrap has taught us is that physical theories like the 2d, 3d Ising models, and the Wilson-Fisher fixed points in fractional dimensions lie at the kinks in allowed spaces of theories [13]. Multiple correlators further constrain these allowed regions down to islands (which we do not consider here). In the same spirit, it is indeed quite striking that the standard model values appear to lie at a kink in the space of allowed S-matrices on the Alder-zeros plane. But what is so unique about the standard model? More generally, what is so special about the models which describe linearity in the resonances? To examine these questions, we will use two different observables: (a) Averaged total scattering cross-section for $\pi^0\pi^0$ and $\pi^+\pi^-$ and (b) Entanglement entropy in isospin space using entanglement power.

## 4.1 Averaged total scattering cross section

Let us consider the total scattering cross section for the process $AB \to$ anything which arises from the optical theorem. Specifically we compute the averaged total cross section given by

$$\bar{\sigma}(s_0) = \frac{1}{2s_{cut}} \int_4^{s_{cut}} ds \sqrt{\frac{s-4}{s}} \, \mathrm{Im}\left(M^{AB \to AB}(s, t = 0)\right), \tag{4.1}$$

with $s_{cut}$ denoting the cut-off $s$ up to which the average is considered. We will consider $\pi^0\pi^0$ and $\pi^+\pi^-$ total scattering cross-sections.

What we find is that $\bar{\sigma}$ dramatically decreases near **A**–see fig.(14). $\partial_{s_0}\bar{\sigma}$ at the point **A** is

the minimum amongst all boundary S-matrices. The location of the $\partial_{s_0}\bar{\sigma}$ minimum does not alter significantly with $s_{cut}$ while the $\bar{\sigma}^{\pi^0\pi^0}$ minimum shifts to $s_0 \approx 0.60$ with higher $s_{cut}$. Furthermore, other reactions like $\pi^+\pi^0 \to \pi^+\pi^0$ lead to the same conclusion for $\partial_{s_0}\bar{\sigma}$. This dramatic drop in the total cross-section is reminiscent of "operator decoupling" in the conformal bootstrap which causes the kink there [13]. It is tempting to conjecture that a similar phenomena is at play here and may in fact pave the road for a non-perturbative understanding of the pion scattering problem in the standard model.

It is also worth noting that the lower boundary intersection point **C** also shows a similar sharp drop, although smaller than **A**. The fact that Regge behaviour, intersection of the LS line, and this drop in $\partial_{s_0}\bar{\sigma}$ occur around the same region seems quite remarkable.

## 4.2   Entanglement Power

In our previous work, [8], following [14] we had considered a quantity called Entanglement Power ($\mathcal{E}$) and had initiated its investigation in the context of pion scattering. Starting from an arbitrary initial state, we define the final state in a specific manner (details given in Appendix E) through the S-matrix. It has the following form,

$$\mathcal{E} = 1 - \int \frac{d\Omega_1}{4\pi}\frac{d\Omega_2}{4\pi}\text{tr}_1[\bar{\rho}_1^2], \quad d\Omega_i := \sin\theta_i d\theta_i d\phi_i. \tag{4.2}$$

Here $\bar{\rho}_1$ is the reduced density matrix obtained after averaging over the isospins of the incoming states and tracing out one of the final state particles. For a $d$-dimensional spin-space, $\mathcal{E}$ is bounded from above by $1 - 1/d = 2/3$ [15]–this provides a nontrivial check for our calculations. $\mathcal{E}$ near threshold $s \approx 4$ has a complicated form as shown in Appendix E, but it can be checked that $\mathcal{E} \leq 2/3$ and occurs for $a_0^{(2)} = 0$. Near threshold, we also find $\mathcal{E} \gtrsim 0.14$. For higher values of $s$, the lower bound can decrease (we have not found an absolute lower bound). In order to define an analogue of the averaged scattering cross section, we shall consider an averaged entanglement power,

$$\bar{\mathcal{E}} = \frac{1}{s_{cut}} \int_4^{s_{cut}} \mathcal{E}ds \tag{4.3}$$

Figures (3.a) and (3.b) show variation of $\mathcal{E}$ with $s_0$. $s_{cut} = 375$ is chosen in order to include the contribution of all experimentally known resonances. The figures suggest that there is a correlation between the sharp decrease in $\bar{\sigma}$ with decrease in $\bar{\mathcal{E}}$. The joint decrease in $\bar{\sigma}$ and $\bar{\mathcal{E}}$ selects out regions **A,B,C,D** as special.

## 4.3   Selection rules

Our findings suggest the following selection criteria which pick out S-matrices describing linear Regge trajectories:

1. $\bar{\sigma}$ exhibits a minimum,–see fig.(3.a), (3.b), as well as the discussion in Appendix E. Mandelstam [16] pointed out that linear Regge trajectory with associated narrow widths, can also be expected to be correlated with low scattering cross sections. Our findings render support to this expectation.

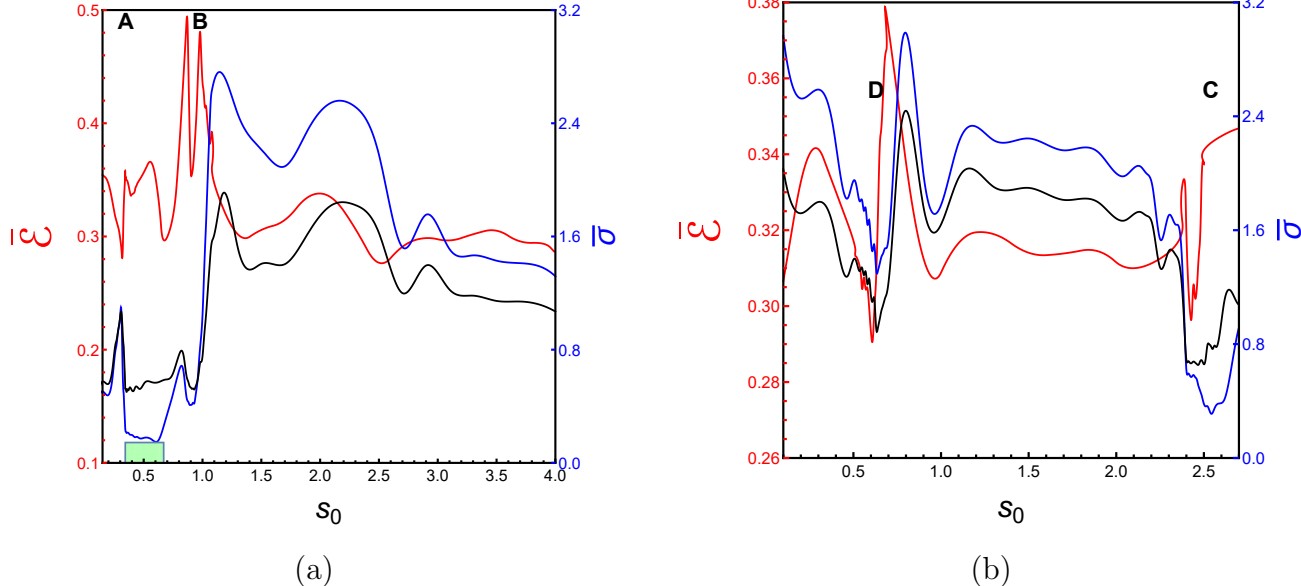

Figure 3: (a) Variation of $\bar{\mathcal{E}}$ and $\bar{\sigma}$ with $s_0$ for $s_{cut} = 375$ for upper boundary. (b) Variation of $\bar{\mathcal{E}}$ and $\bar{\sigma}$ with $s_0$ for the lower boundary for $s_{cut} = 375$. Blue line is for $\bar{\sigma}^{\pi^0\pi^0}$, black for $\bar{\sigma}^{\pi^+\pi^-}$ and red for $\bar{\mathcal{E}}$. The green bar indicates linear Regge trajectory with $R^2 > 0.93$.

2. $\bar{\mathcal{E}}$ exhibits a minimum. While one may have expected that a minimum in the total cross-section will be correlated with reduction in entanglement, fig.(3) makes it clear that the relation is more subtle. This is not entirely unexpected since $\mathcal{E}$ involves an averaging in isospin space as well, while the total scattering cross-section involves averaging over $s$ only. It is clear however, that onset of linear Regge behaviour happens when both $\bar{\sigma}$ and $\bar{\mathcal{E}}$ are minimized.

3. The sharpest decrease in $\bar{\sigma}$ occurs near **A** which remarkably is the location of the standard model, as witnessed in fig. (14). On the upper boundary near $s_0 \approx 0.9$, all three quantities show a minimum; however the even and odd spin resonances lie on straight lines with different slopes. This indicates that the minimization of $\bar{\sigma}$, $\bar{\mathcal{E}}$ criteria are not sharp enough to distinguish when the even, odd slopes are the same. The minimum in $\partial_{s_0}\bar{\sigma}$ is a better indicator of this feature.

Applying these criteria select out S-matrices in regions **A**, **C** as special. They also have S and P-wave scattering lengths compatible with experiments[5].

## 5  Diving into the allowed region

Inspired by our observations in the previous section, we consider a method to investigate the possibility of using the selection rules (as described in 4.3) within the river. We are required to extremize some quantity to determine a unique S-matrix for any point in the allowed space. The

---

[5]We refer to the experimental values quoted in [9], namely $a_0^{(0)} = 0.2220 \pm 0.0215_{tot}, a_0^{(2)} = -0.0432 \pm 0.0148_{tot}$. Note that there are more stringent values quoted in [9] which were used in [6] but these need inputs from analyticity and $\chi PT$ which we will not use. For the P-wave we will use the older result quoted in [6], $a_1^{(1)} = 0.038 \pm 0.002$.

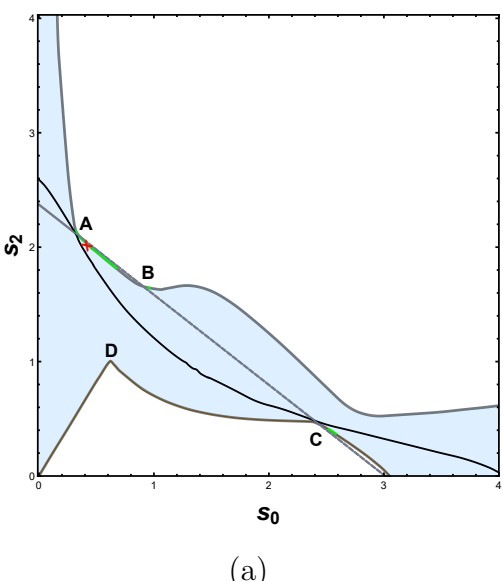

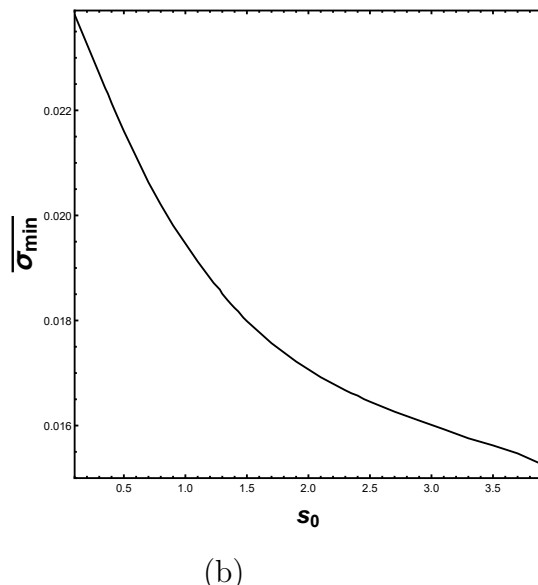

(a)                                                (b)

Figure 4: (a) The black curve within the river river gives the $s_2$ with minimum average cross section for each $s_0$. Note that the curve passes very close to regions A and C described earlier (b) The value of the corresponding minimum as a function of $s_0$ for $N_{max} = 14$ and $L_{max} = 19$. The global minimum average cross section appears to be at $s_0 = 4$

averaged total cross section is the best candidate for this minimization as it is linear in the ansatz parameters.

Therefore we minimize the $\pi^0 + \pi^0 \to \pi^0 + \pi^0$ averaged total cross section for each allowed point $s_0, s_2$; namely we use eq.(4.1) with $s_{cut} = 375$. Now, for each $s_0$, we determine the $s_2$ which has the minimum $\bar{\sigma}$. This generates a curve of minimum $s_2$ points. Plotting vs $s_0$, we get the minimum curve given in figure 4. Remarkably, this curve passes very close to both A and C. *We also repeated the analysis after removing the S and D wave inequalities (which produced the river) and observed no change to the minimum curve.* Furthermore, all the S-matrices along the curve show Regge behaviour for even spins. This validates our observation motivated by [16] that minimizing $\bar{\sigma}$ will lead to Regge behaviour. Furthermore, in fig.(6), we plot the entanglement power along this curve and observe local minimum near A and C. Convergence properties with $N_{max}$ and $L_{max}$ are provided in appendix D.

We can also perform hypothesis testing following [8] with $\chi PT$ by calculating averaged $\overline{S_R(\rho_1\|\rho_2)}$,

$$
\begin{aligned}
\overline{S_R(\rho_1\|\rho_2)} &= \int_4^{s_{cut}} ds \int_{-1}^1 dx \, \mathcal{P}_g^{boot}(x) \ln\left(\frac{\mathcal{P}_g^{boot}(x)}{\mathcal{P}_g^{\chi PT}(x)}\right) \quad \text{and} \\
\mathcal{P}_g(x) &= \frac{g(x)\,|\mathcal{M}(s,x)|^2}{\int_{-1}^1 dx\, g(x)\,|\mathcal{M}(s,x)|^2}, \quad g(x) = \frac{1}{2\sqrt{2}\sigma} e^{-\frac{(x-y)^2}{4\sigma}} .
\end{aligned}
\tag{5.1}
$$

where $\rho_1$ comes from bootstrap and $\rho_2$ comes from $\chi PT$, $y$ is chosen to be 0.01 and $\sigma$ is chosen to be $10^{-6}$ . Hypothesis testing results do not depend on $y$. As shown in 6, we take a maximum $s_{cut}$ of 20 since the validity of $\chi PT$ decreases at higher $s$. Very interestingly, we observe that the S-matrices near region A show the minimum deviation from $\chi PT$. Thus hypothesis testing(w.r.t $\chi PT$) favours region A of the minimum curve.

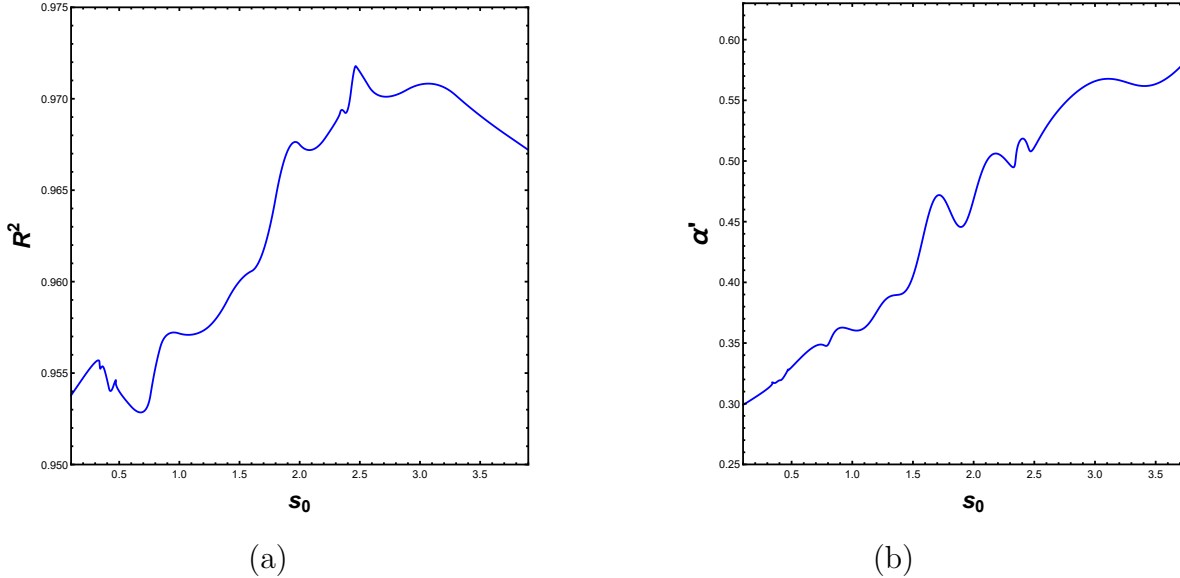

(a)                                             (b)

Figure 5: (a) Variation of best-fit $R^2$ for the S-matrices on the minimum line. Only even spins are considered. An average $R^2 > 0.95$ shows that all such S-matrices show good linearity (b) Variation of best-fit slope($\alpha'$) of the S-matrices as a function of $s_0$.

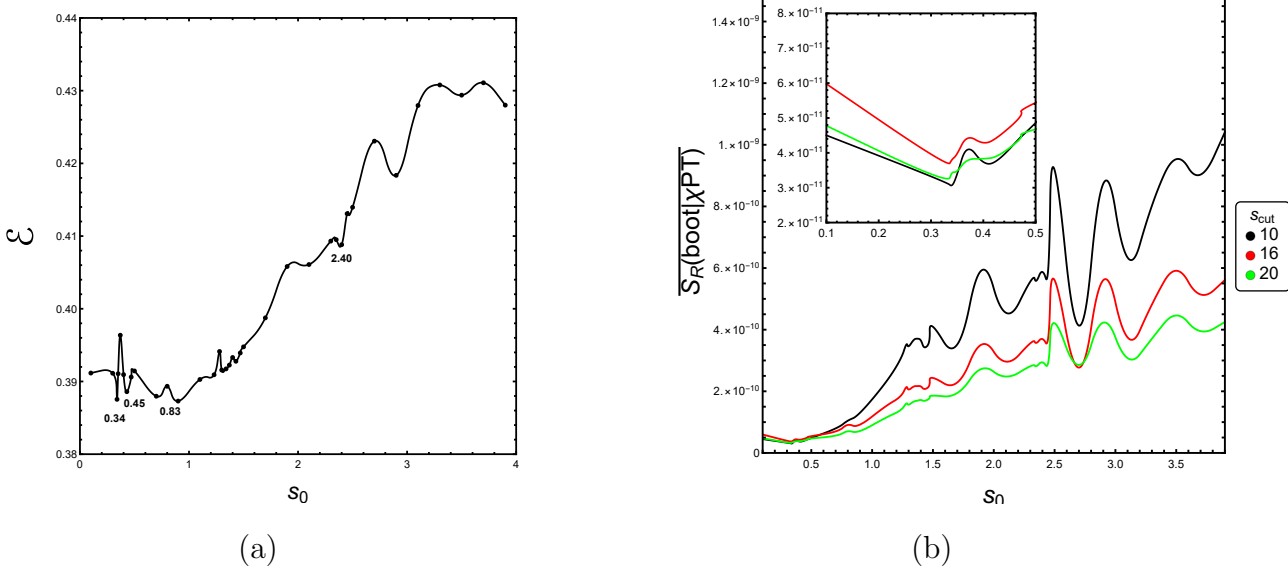

(a)                                             (b)

Figure 6: (a) Variation of $\mathcal{E}$ for the S-matrices on the minimum line. (b) Variation of averaged Relative Entropy ($\overline{S_R(\rho_1|\rho_2)}$) of the S-matrices as a function of $s_0$. A global minimum is obtained near region A.

# 6 Future directions

The minimization of entanglement power being correlated with interesting physical theories has been alluded to before in [14] and in a semi-classical context of black hole physics in [17]. Here we find a remarkable correlation between the minimization of total scattering cross-section, entanglement power, and emerging linear Regge behaviour. The minimization of entanglement is consistent with emerging classicality–see, for instance, [18]. One may expect that for effective field theory description to be valid, such a reduction in entanglement must happen. For a minimization in entanglement, one may also expect that interaction will be reduced, a fact corroborated by the total scattering cross-sections' behaviour. This chain of arguments supports our findings in this paper, and it will be very worthwhile to investigate further[6].

We found two potentially interesting regions (**A** and **C** in fig.(1)), which satisfy linearity in resonances and exhibit the experimental S and P-wave scattering lengths. Which of these regions then describes the real world? We do not have a definitive answer to this fascinating question[7], and we leave it to future work to settle this. Our hypothesis testing using quantum relative entropy does suggest that these S-matrices are "close" to one another in the manner discussed in [8]. One parting comment is that the absolute value for the interaction range $\left|b_0^{(2)}\right|$ for the **C** region is an order of magnitude bigger than **A**, which itself is in the ball-park that is predicted by $\chi PT$. May be, only experiments will settle this issue in the future.

From a practical point of view, to investigate the physics considered in this paper further, it would be desirable to have better and faster numerical approaches, perhaps building on the recent proposals in [20, 21]; for instance, using the current methods, the decay widths appear to be quite sensitive to $N_{max}$. Furthermore, it may also be beneficial to consider a better starting point inspired by the narrow resonance approximation. The current ansatz being employed may be too restrictive to study higher energies–for instance, there is little hope for exploring the Froissart bound using present numerics. A more ambitious program of trying to connect with string theory (even a less ambitious question of probing daughter trajectories) will need such a development. It may also be a fruitful exercise to correlate the observations in this paper in the large $N$ limit using the AdS/CFT correspondence.

# Acknowledgments

We thank Andrea Guerrieri and Balt van Rees for correspondence on the bootstrap numerics and, Parthiv Haldar and Prithish Sinha for collaboration in [8]. Encouraging comments from Sougato Bose, Giancarlo D'Ambrosio, Rohini Godbole, Rajesh Gopakumar, David Gross, Anupam Mazumdar, Joao Penedones as well as discussions with Abhijit Gadde, Sachin Jain, Shiraz Minwalla and Sandip Trivedi are gratefully acknowledged. Special thanks are due to Parthiv Haldar and Apratim Kaviraj for comments on the draft. We thank Urbasi Sinha's lab in RRI for providing us access to a workstation during the course of this work. We are especially grateful to ICTS,

---

[6] For instance, it may be an interesting exercise to consider the massless limit as in [19] and calculate the observables in this paper.

[7] While fig. (8) suggests that **A** matches better with experiments, we cannot definitively rule out **C** just yet–see the discussion in appendix D.

Bangalore (Rajesh Gopakumar, Samriddhi S. Ray and Hemanta Kumar) for enabling us access to the ICTS computing clusters (Mario and Tetris, our virtual brothers) which enabled us to perform the enormously time consuming SDPB simulations (we really hope easier methods are available soon!). Special thanks to Hemanta Kumar who performed a heroic source installation of SDPB on the clusters and who very promptly answered all our laymen queries for using the clusters.

# A    S-matrix bootstrap review

Here we briefly review the S-matrix bootstrap describing $2 \to 2$ scattering of pions as considered in [6, 8], which built on [7]. Let the initial state and final state be $|p_1, a; p_2, b\rangle$ and $|q_1, c; q_2, d\rangle$ respectively, where $a$, $b$, $c$ and $d$ are $O(3)$ group indices. The S-matrix can be defined as

$$\langle q_1, a; q_2, b|S|p_1, a; p_2, b\rangle = \mathbf{1} + i \, \delta^4(p_1 + p_2 - q_1 - q_2) \, \mathcal{M}_{a\,b}^{c\,d}(s, t, u), \tag{A.1}$$

where $s, t, u$ are the usual Mandelstam variables. $\mathcal{M}_{a\,b}^{c\,d}(s, t, u)$ has $O(3)$ symmetry and hence can be expanded as

$$\begin{aligned}
\mathcal{M}_{a\,b}^{c\,d}(s, t, u) &= A(s|t, u)\delta_{a\,b}\delta^{c\,d} + A(t|u, s)\,\delta_{a\,c}\,\delta_{b\,d} + A(u|s, t)\,\delta_{a\,d}\,\delta_{b\,c}, \\
&= (3A(s|t, u) + A(t|u, s) + A(u|s, t))\,\mathbb{P}_0 + (A(t|u, s) - A(u|s, t))\,\mathbb{P}_1 \\
&\quad + (A(t|u, s) + A(u|s, t))\,\mathbb{P}_2.
\end{aligned} \tag{A.2}$$

$\mathbb{P}_I$ are the 3 projectors of the $O(3)$ group channels defined as

$$\mathbb{P}_{\text{sing}} = \mathbb{P}_0 = \frac{1}{3}\delta_{ab}\delta^{cd}, \quad \mathbb{P}_{\text{anti}} = \mathbb{P}_1 = \frac{1}{2}(\delta_a^c\delta_b^d - \delta_a^d\delta_b^c), \quad \mathbb{P}_{\text{sym}} = \mathbb{P}_2 = \frac{1}{2}(\delta_a^c\delta_b^d + \delta_a^d\delta_b^c - \frac{2}{3}\delta_{ab}\delta^{cd}). \tag{A.3}$$

Crossing symmetry constraints $A(s|t, u)$ to follow $A(s|t, u) = A(s|u, t)$. Next, the partial wave expansion is given by

$$\mathcal{M}(s, t, u) = 16\, i\, \pi \frac{\sqrt{s}}{\sqrt{s-4}} \sum_{I=0,1,2} \mathbb{P}_I \sum_{\ell=0}^{\infty} (2\ell + 1)\left(1 - S_\ell^{(I)}(s)\right) P_\ell\left(x = \frac{u-t}{u+t}\right). \tag{A.4}$$

We use the following crossing symmetric ansatz for $A(s|t, u)$

$$A(s|t, u) = \sum_{n \leq m}^{\infty} a_{nm}\,(\eta_t^m \eta_u^n + \eta_t^n \eta_u^m) + \sum_{n,m}^{\infty} b_{nm}\,(\eta_t^m + \eta_u^m)\,\eta_s^n, \tag{A.5}$$

where $\eta_s = \dfrac{\left(\sqrt{4 - \frac{4}{3}} - \sqrt{4-s}\right)}{\left(\sqrt{4 - \frac{4}{3}} + \sqrt{4-s}\right)}$, and similarly for $\eta_t$, $\eta_u$. Following [6], we truncate to $N_{max}$ and impose unitarity for $L_{max}$ partial waves through $\left|S_\ell^{(I)}(s)\right|^2 \leq 1$ for a grid of s-values. We can also define

$$f_\ell^{(I)}(s) = \sqrt{\frac{s}{s-4}}\frac{S_\ell^{(I)}(s) - 1}{2i}, \tag{A.6}$$

to satisfy the equivalent unitarity condition of $\mathrm{Im}\left(f_\ell^{(I)}(s)\right) \geq 2\sqrt{\frac{s-4}{s}}\left|f_\ell^{(I)}(s)\right|^2$. In terms of $T_\ell^{(I)}(s)$, the scattering lengths $(a_\ell^{(I)})$'s and effective ranges $(b_\ell^{(I)})$'s can be defined as,

$$\mathrm{Re}\left[f_\ell^{(I)}(k)\right] = k^{2\ell}\left[a_\ell^{(I)} + b_\ell^{(I)}k^2 + \mathcal{O}(k^2)\right], \quad k = \frac{\sqrt{s-4}}{2}. \tag{A.7}$$

To get unique S-matrices, we extremize a linear combination of above parameters ($a_{nm}$ and $b_{nm}$), under the constraint of unitarity, using SDPB [22], similar to [6]. To specialize for pions, we include the $\rho$-resonance. Resonances will be further described in Appendix B. The $\rho$-resonance is imposed as,

$$S_1^1\left(m_\rho^2\right) = 0, \quad m_\rho = 5.5 + i\,0.5. \tag{A.8}$$

The sign of the imaginary part is such that it corresponds to a zero in the physical sheet. For bootstrap, we shall consider the Adler zeros in singlet and symmetric channel using,

$$S_0^{(0)}(s_0) = 1 \quad \text{and} \quad S_0^{(2)}(s_2) = 1. \tag{A.9}$$

Tree-level $\chi PT$ predicts $s_0 = 0.5$, $s_2 = 2$, one-loop has zeroes at $s_0 = 0.437$, $s_2 = 2.003$, while the two-loop values [11] are $s_0 = 0.4195$, $s_2 = 2.008$. The location of the zero in the $(s_0, s_2)$ plane appears to move towards the "kink" at **A** located at $s_0 \approx 0.34$, $s_2 \approx 2.1$.

To proceed, we first determine which of these pairs of Adler zeros are allowed by unitarity and the $\rho$-resonance. This is checked by imposing $T_0^{(0)}(s_0) = 0$ for some $s_0 \in (0,4)$ and checking the sign of $\mathrm{Max}\left(T_0^{(2)}(s_2)\right)$ and $\mathrm{Min}\left(T_0^{(2)}(s_2)\right)$. If the maxima is positive and minima is negative, the point $(s_0, s_2)$ in the Adler zero plane is allowed. Repeating this for different $s_0's$ and $s_2's$ we get the pion lake in [6].

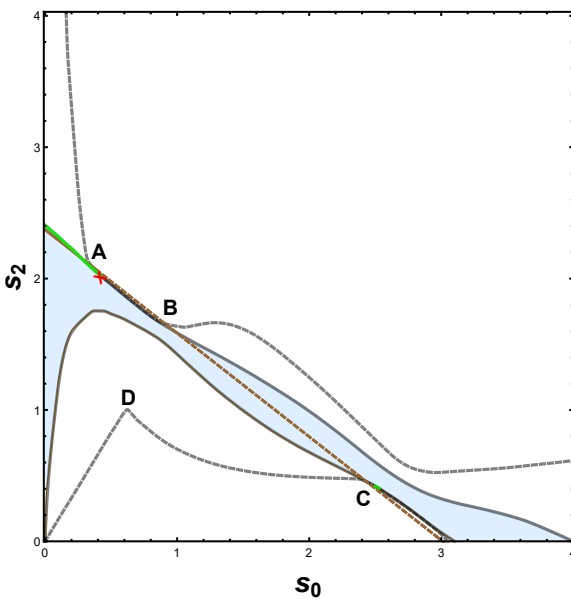

Figure 7: Experimental "Peninsula" inside the river (indicated by grey dashed line). This serves as an indicator as to the points where we can expect experimental scattering lengths. Green indicates $R^2 > 0.9$.

Now since the disallowed region of pion lake is very small, one method of increasing the disallowed region is to impose more experimental constraints. The scattering lengths $a_0^{(0)}, a_0^{(2)}$ and $a_1^{(1)}$

can be constrained to be within the experimental values [6,9] $0.2220\pm0.0215, -0.0432\pm0.0148, 0.038\pm$ $0.002$ respectively. Imposing these values, in addition to the $\rho$-resonance and unitarity, gives us the pion Peninsula, as shown in fig. 4. Note that the experimental values we use here are the weaker ones quoted in [9] which do not use any analyticity or $\chi PT$ inputs. As a result, the peninsula we plot below is somewhat larger than the one in [6].

# B  Detecting resonances

In this section, we will outline our strategy to locate the resonances. Recent discussions include [23, 24] in the context of 2d-bootstrap. However, our approach will be a more approximate one, mimicking what happens in an experiment, following the discussions in [25, 26]. Resonances occur as poles of the S-matrix, with non-zero imaginary part (since the wave function must decay with time). The complex pole is at $s = s_r = m_A^2 - im_A\Gamma_{\text{total}}$, where $\Gamma$ is the decay width and this pole must be in the second sheet as we will review below.

Subsequently, the partial waves can inherit these poles after projection for isospins $I = 0, 1, 2$ and angular momentum $\ell$. Let us call partial wave on physical sheet as $S_\ell^{(I)}(s)$ and on the second sheet as $R_\ell^{(I)}(s)$. Now, we know that the threshold due to $\Pi(-s)$ is actually responsible for square-root branch cut starting at $s = 4$. It is a single square root type branch cut in the elastic region which connects two sheets (more complicated for multiple branch cut systems). Now we write down the elastic-unitarity condition as

$$\lim_{\epsilon\to0^+} S_\ell^{(I)}(s + i\epsilon)S_\ell^{(I)}(s - i\epsilon) = 1.\tag{B.1}$$

Since, in the elastic range of the branch-cut we must have that $\lim_{\epsilon\to0^+} R_\ell^{(I)}(s + i\epsilon) = S_\ell^{(I)}(s - i\epsilon)$, hence,

$$\lim_{\epsilon\to0^+} S_\ell^{(I)}(s + i\epsilon)R_\ell^{(I)}(s + i\epsilon) = 1.\tag{B.2}$$

This is a product of two analytic functions. Now, if we map both the sheets or part of both the sheets into one, connected through the elastic region branch-cut, the product of these analytic function will remain 1 as we extend to the whole domain containing this elastic region. If there was a pole at $s \approx m_A^2 - im_A\Gamma_{\text{total}}$ in the second sheet and $m_A^2$ is smaller than the inelastic threshold, then, eq.(B.2) will imply a a zero at $s \approx m_A^2 - im_A\Gamma_{\text{total}}$ in the physical sheet. Therefore, from Schwartz reflection principle,

$$S_\ell^{(I)}(m_A^2 + im_A\Gamma_{\text{total}}) = (S_\ell^{(I)}(m_A^2 - im_A\Gamma_{\text{total}}))^* = 0\tag{B.3}$$

This is precisely the resonance condition being used on the physical sheet.

## B.1  Breit-Wigner form

Here we will briefly summarize the Breit-Wigner form for resonances. Assuming a well separated resonance at $s = m_\ell^2 - im_\ell\Gamma$ for the $\ell^{th}$ partial wave, we will have the form

$$f_\ell(s) = \frac{g_\ell(s)}{s - (m_\ell^2 - im_\ell\Gamma)}, \quad g_\ell(s) \in \mathbb{R}.\tag{B.4}$$

Now, assuming that $\Gamma \ll m_\ell$, *i.e.* a small enough decay rate, we can analytically continue this form from below the branch cut in the second sheet onto the branch cut. Next,when $s$ is real and $s > 4$, we can impose unitarity, or even stronger, elastic unitarity. Thus we have that

$$|S_\ell(s)|^2 = 1 \implies g_\ell(s) = -m_\ell \Gamma \sqrt{\frac{s}{s-4}} . \tag{B.5}$$

This gives us that

$$S_\ell(s) = \frac{(s - m_\ell^2) - i m_\ell \Gamma}{(s - m_\ell^2) + i m_\ell \Gamma} . \tag{B.6}$$

This form has the required zero in the first sheet, if we continue extending further. Note that this is a consequence of strict elastic unitarity.

Next, we see that in eq.(B.6), when we scan the real axis (which is what we have access to experimentally), our partial wave will behave as

$$S_\ell(s) \xrightarrow{s \to m_\ell^2} -1 + 2\left(\frac{s - m_\ell^2}{m_\ell \Gamma}\right)^2 - 2i\left(\frac{s - m_\ell^2}{m_\ell \Gamma}\right) . \tag{B.7}$$

So we have that near the real part of the resonance, the amplitude will tend to $-1$, or equivalently, the phase tends to $\pi$. It is the latter which is of use to us since in cases when elastic unitarity is not valid, we can instead define the resonance through a sudden change in phase! Another alternate definition can be motivated by looking at the form of $f_\ell(s)$ found from eq.(B.6) which leads to

$$|f_\ell(s)|^2 = 2\frac{s}{s-4}\frac{m_\ell \Gamma}{(s - m_\ell^2)^2 + m_\ell^2 \Gamma^2} . \tag{B.8}$$

Hence, we see that $f_\ell(s)$ and more generally $|f_\ell(s)/S_\ell(s)|^2$ has a peak at $s = m_\ell^2$ and this can be used as a much more general definition of a resonance. Using this strategy we find fig.(8). For a further check of validity, see appendix D. We have observed that unlike the peak locations, the widths are not in good agreement with experiments and are sensitive to $N_{max}$ and we will refrain from presenting them.

## B.2   Linearity of even and odd trajectories

Here we give plots where linearity emerges for odd and even spins in fig.(9). In fig.(10) we plot the statistical coefficient of determination $R^2$. Only in region **A** and to a lesser extend in region **C** do the odd and even spins line up together.

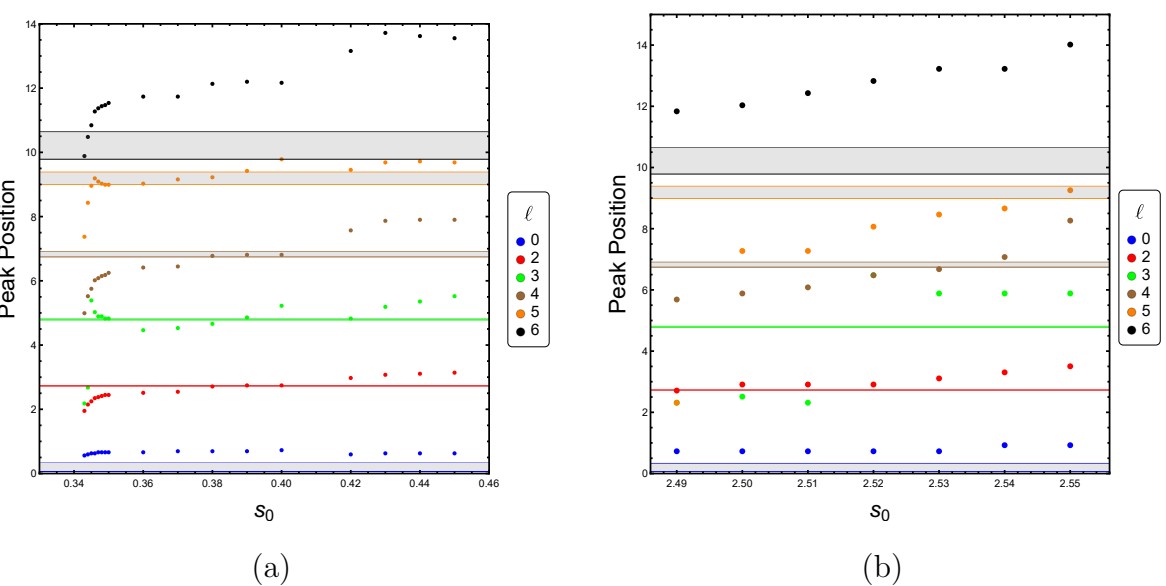

Figure 8: (a) Position of peaks vs $s_0$ near the kink **A** and (b) Position of peaks vs $s_0$ near the kink **C**. These peaks were calculated for $N_{max} = 16$ and $L_{max} = 19$. The bands of a particular color indicate the experimental range of resonances associated with that spin.

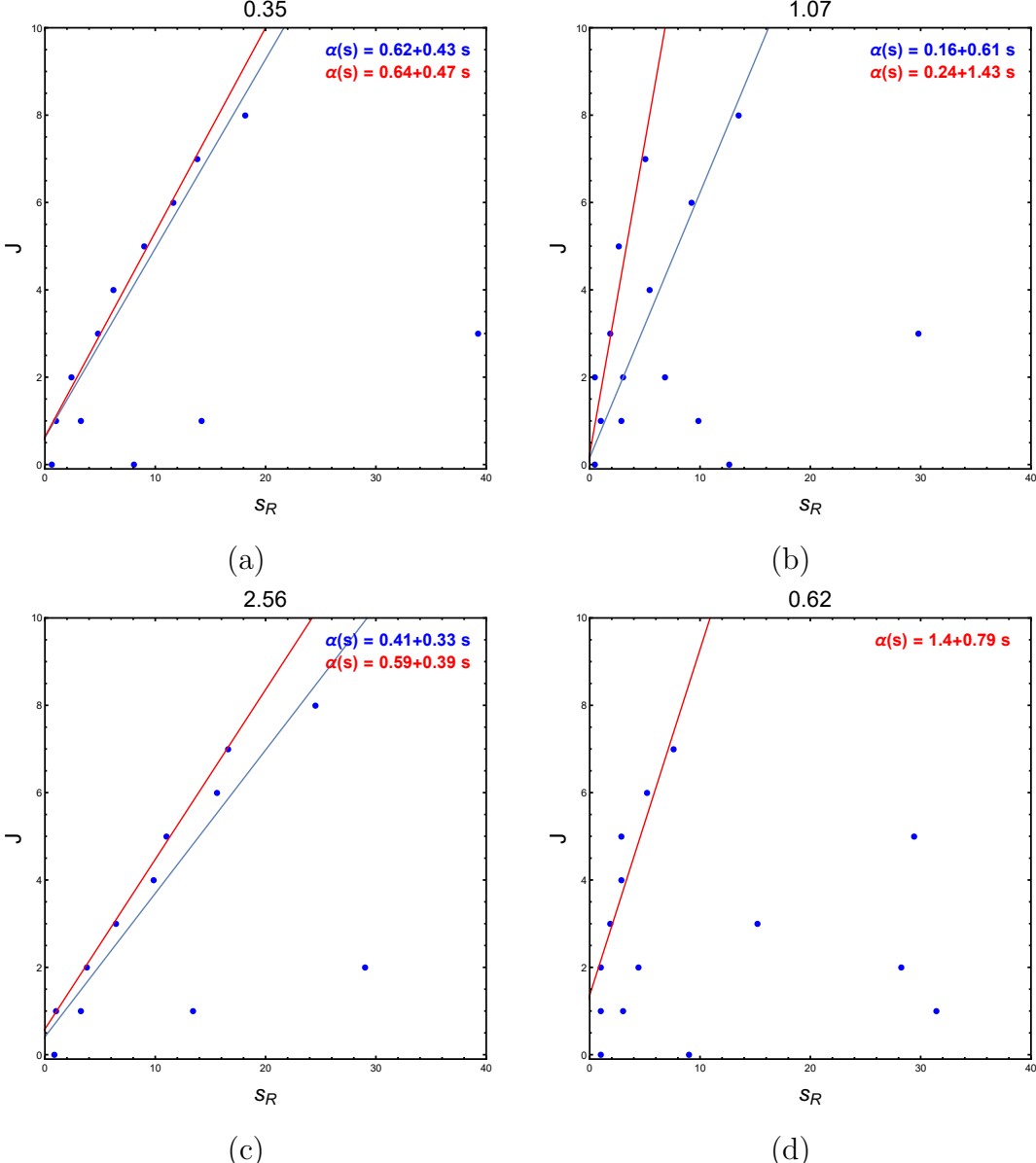

Figure 9: (a) Region **A**: Red lines give best fit odd trajectories and blue lines describe best fit even trajectories (equations mentioned in the inset). Both even and odd lines are close together. (b) Region **B**: Even and odd lines somewhat separated (c) Region **C**: Even and odd lines closer than region B but farther than region A and (d) Region **D**: $\ell = 8$ missing. Some daughter resonances can be observed in all regions.

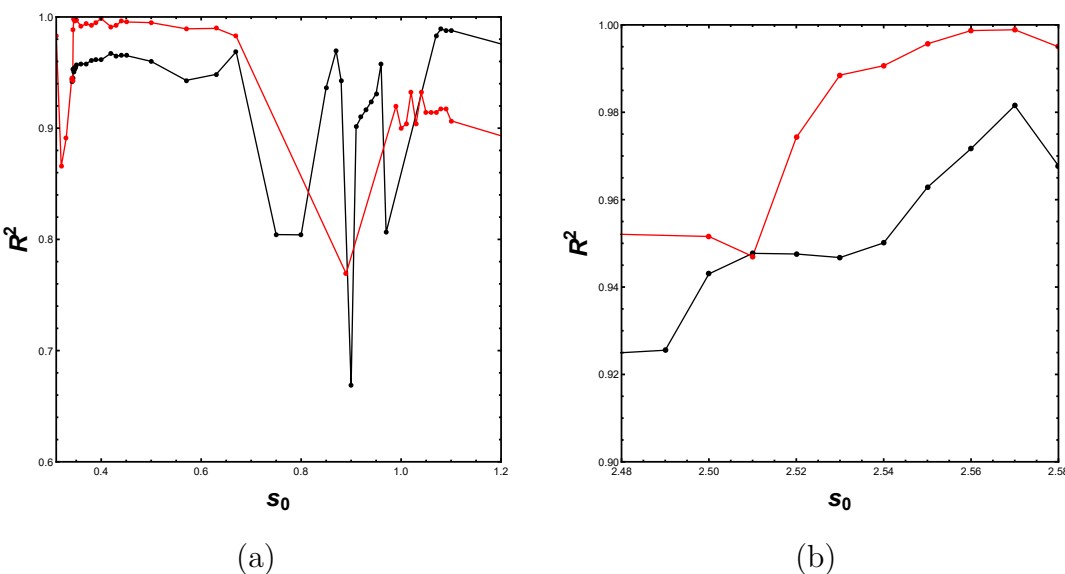

(a)                                                              (b)

Figure 10: (a) Region **A**: Plot of $R^2$ vs $s_0$ of even(Black) and odd(red) trajectories for upper boundary near Region A. We have good linearity when $R^2$ (coefficient of determination) is closer to 1. (b) Region **B**: Plot of $R^2$ vs $s_0$ of even(Black) and odd(Red) trajectories for upper boundary near Region C

# C  Lovelace-Shapiro model in the plane of Adler zeros

Here, we will briefly discuss the Lovelace-Shapiro (LS) model. We start with the form of the amplitude in the LS model as [3, 10, 27]

$$A^{(0)}(s,t,u) = \frac{3}{2}(A(s,t)+A(s,u)) - \frac{1}{2}A(t,u), \quad A^{(1)}(s,t,u) = A(s,t) - A(s,u), \quad A^{(2)}(s,t,u) = A(t,u),$$
(C.1)

with

$$A(s,t) = C_4 \frac{\Gamma(1-\alpha(s))\Gamma(1-\alpha(t))}{\Gamma(1-\alpha(s)-\alpha(t))} .$$
(C.2)

Here, $C_4$ is a normalization constant and $\alpha(s)$ is the normalized, linear Regge trajectory of the $\rho$-meson (or equivalently the $\rho$ resonance) such that

$$\alpha(s) = \alpha_0 + \alpha' s, \quad \alpha(m_\rho^2) = 1 .$$
(C.3)

Now it is usual to demand that the Regge trajectory is fixed by demanding an Adler zero when one of the external momenta goes to 0. This can be easily implemented using the poles of the Gamma function. In other words, we simply demand a pole of the Gamma function in the denominator wherever we want a zero of the amplitude. Now, when one of the external momenta goes to 0 (lets choose $p_1 \to 0$ $w.l.o.g$), we have that $s = (p_1+p_2)^2 = p_2^2 = m_\pi^2$, $t = (p_1-p_3)^2 = p_3^2 = m_\pi^2$. Hence, we have an Adler zero at $s = t = m_\pi^2$ and therefore, the Gamma function in the denominator must have a pole there. We choose the first pole of the Gamma function for this purpose, $2\alpha(m_\pi^2) = 1$. This, along with the normalization of $\alpha(m_\rho^2) = 1$ is enough to fix the trajectory as

$$\alpha_0 = \frac{m_\rho^2 - 2m_\pi^2}{2(m_\rho^2 - m_\pi^2)}, \quad \alpha' = \frac{1}{2(m_\rho^2 - m_\pi^2)}$$
(C.4)

It can be shown that the above is equivalent to demanding that $s_2 = 2(m_\pi^2)$ in our Adler zero language. This is so as $s_2$ will be the zero of $A_0^{(2)}(s)$ such that

$$A_0^{(2)}(s) = \frac{1}{2} \int_{-1}^{1} dx P_0(x) A^{(2)}(s, t(x), u(x)) ,$$
(C.5)

with $t(x) = -\frac{1}{2}(s-4m_\pi^2)(1-x)$, $u(x) = -\frac{1}{2}(s-4m_\pi^2)(1+x)$ being the Mandelstam variables in terms of the scattering angle $\cos(\theta) = x$.

Now, we observe that

$$A^{(2)}(s, t(x), u(x)) = \frac{\Gamma(1-\alpha(t(x)))\Gamma(1-\alpha(u(x)))}{\Gamma(1-\alpha(t(x))-\alpha(u(x)))} .$$
(C.6)

So, we can see that the numerator is a complex function of $x$. However, the denominator is actually independent of $x$ as $1 - \alpha(t(x)) - \alpha(u(x)) = 1 - 2\alpha\left(\frac{4m_\pi^2-s}{2}\right)$.

Therefore, the denominator can be taken out of the integral in eq.(C.5) directly. This leads to the partial wave $A_0^{(2)}(s)$ inheriting the pole structure of $\Gamma\left(1 - 2\alpha\left(\frac{4m_\pi^2-s}{2}\right)\right)$ in the denominator. Equivalently, we must have that $A_0^{(2)}(s_2) = 0$ should imply that $1 - 2\alpha\left(\frac{4m_\pi^2-s_2}{2}\right) = 0$. This leads to $s_2 = 2m_\pi^2$.

Now, we want to generalize this Regge trajectory such that we do *not* demand a specific Adler zero. Instead, we consider the Adler zeros to be free parameters which can take values between $(0,4)$ (back in units of $m_\pi^2 = 1$). This is equivalent to the procedure where under the normalization $\alpha(m_\rho^2) = 1$, we scan the $(\alpha_0, \alpha')$ parameter space. While scanning, we calculate the Adler zeroes $(s_0, s_2)$ numerically for each such value of $(\alpha_0, \alpha')$. Upon doing this, we will obtain a curve in the $(s_0, s_2)$ plane. Then, we see that all feasible Adler zeroes can be theoretically parametrized using $\alpha_0$. Therefore, the set of points $(s_0(\alpha_0), s_2(\alpha_0))$ will form a curve in the Adler zero space. What we actually end up observing is that only for a small range of values of the parameter, do the Adler zeros actually exist. Furthermore, when they do exist, they surprisingly form a straight line in the Adler zero space with the approximate formula of $s_2 = 2.37168 - 0.787739\, s_0$. Note that this is remarkably close to the large $\rho$-mass straight line approximation to the lake [6] which is given by $s_2 = 2.4 - 0.8 s_0$ and will pass through the free theory point $(s_0, s_2) = (0.5, 2)$.

While scanning the parameter space of $(\alpha_0, \alpha')$, we first of all observe that the Adler zeroes $s_0, s_2$ do not exist for the majority of the space. For instance, $s_2$ lies in its expected region of $(0,4)$ only for a tiny range of $\alpha_0 \in (0.46, 0.5)$ which is further decreased when considering both the Adler zeroes simultaneously. Overall, in the total LS line, $\alpha_0$ varies approximately in the range of $(0.465, 0.486)$ while the corresponding range of $\alpha'$ (which is fixed from the normalization of $\alpha(m_\rho^2) = 1$) is approximately $\alpha' \in (0.0177, 0.0170)$–which in units where $m_\rho = 1$ becomes $\alpha' \in (0.535, 0.514)$. Lastly, in the neighborhood of the kink, $\alpha_0 \in (0.48385, 0.48395)$ and $\alpha' \approx 0.516$. Furthermore, we have checked using the arguments in [10] that the models of interest here are *not* unitary.

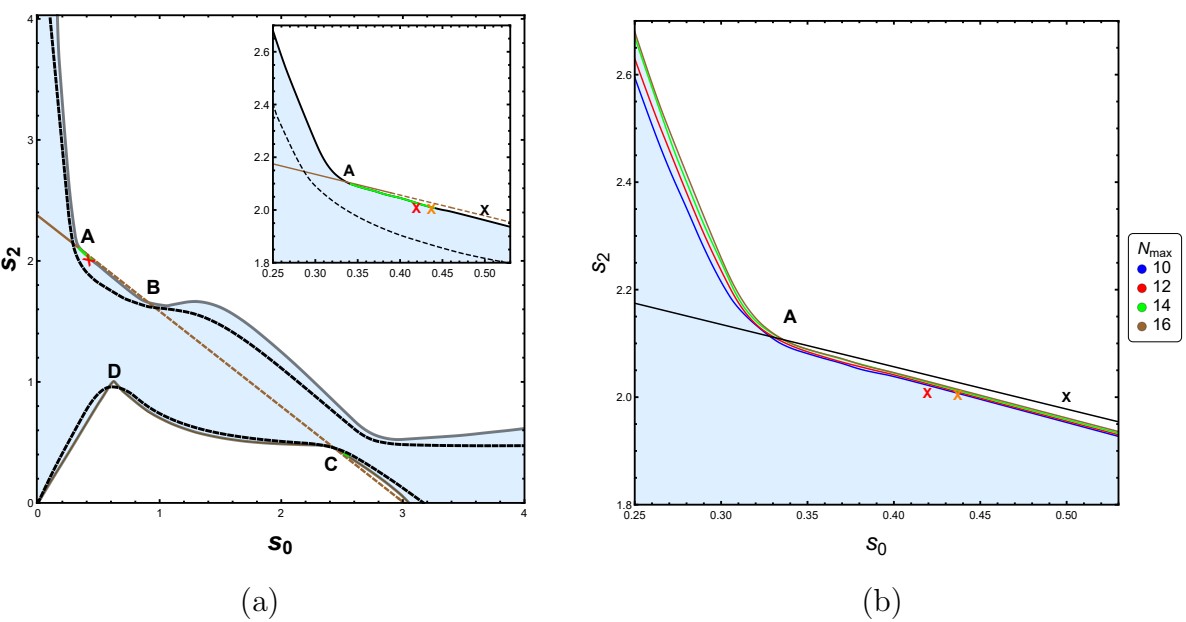

(a)  (b)

Figure 11: (a) This is the river (black dashed) with all resonances imposed at the mean value of the experimental ranges. The values of resonances determined from the S-matrices of this river serve as our benchmark for the method in appendix B. (b) Behaviour near kink for different $N_{max}$. The LS line (black) passes very close to the standard model kink. Also note the transition from the tree-level (black cross) to the 1-loop (orange cross) and finally to the 2-loop (red cross) $\chi PT$ values.

# D  Numerics: Checks

## Determination of Peaks

As described in Appendix B.1, we shall look to find the peaks of $|f_\ell(s)|^2$ to determine the location of the resonances. However the Breit-Wigner form depends on whether elastic unitarity is satisfied or not. Since we cannot (at least not yet) impose elastic unitarity, we check the peaks of $|f_\ell(s)|^2/|S_\ell|^2$ instead.

As an exercise, we also constructed another river (fig 11) by imposing resonances upto $\ell = 6$ at masses [25],

$$m_\sigma = 3.5 - 2\,i, \; m_\rho = 5.5 - 0.5\,i, \; m_{f_2} = 9 - 0.7\,i, \; m_{\rho_3} = 12 - 0.6\,i$$
$$m_{f_4} = 15 - 0.8\,i, \; m_{\rho_5} = 17 - 1.8\,i, \; m_{f_6} = 18 - 1.1\,i \tag{D.1}$$

Apart from $\sigma$, all other resonances gave favourable results as a function of $s_0$ in the sense that the location of the peak did not vary more than a few percent. However, we observed a large variation of the $\sigma$ peak with $s_0$. Nevertheless, since the variation was within the (large) experimental error, we should not dismiss our sigma values of fig.(8).

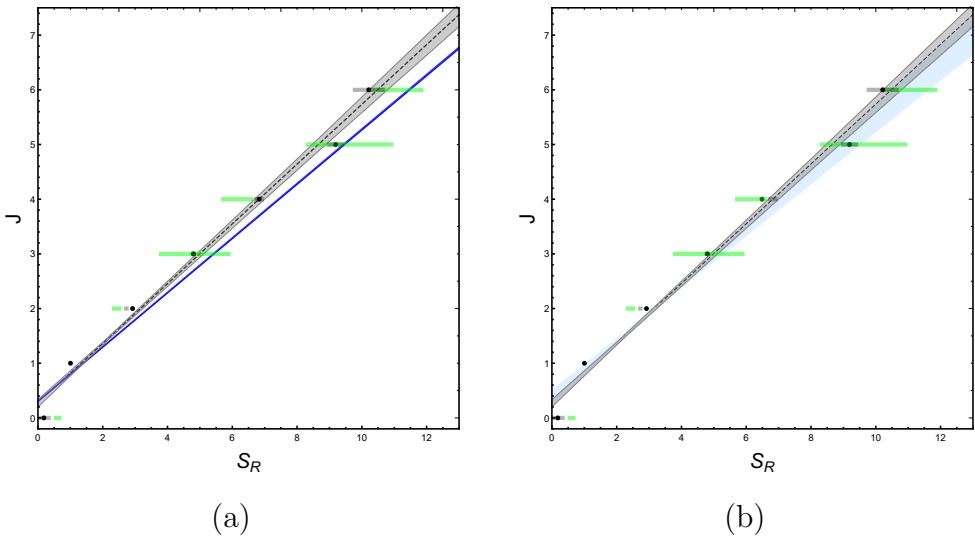

(a)                      (b)

Figure 12: (a) The green bands show the deviation in peak values with different $L_{max} \in [17, 21]$. The Blue band shows the variation in best fit lines. The dark grey bands are the experimental errors of known resonances, the black circles and squares are the experimental $s_R$ of the resonances. The grey dotted line is the experimental best fit and the grey region is variation in experimental best fit (b) The green bands show the deviation in peak values with different $N_{max} \in [14, 20]$.

## Convergence with $L_{max}, N_{max}$

To demonstrate convergence with $L_{max}$, we shall fix the point $s_0 = 0.35$ where we are imposing the Adler zero, and also the maximisation point $s_2 = 2.89$. We shall work with $N_{max} = 16$. As can be seen in fig.(12), we can see good convergence with $L_{max}$. The very small extent of blue region and the green bands indicates small variation.

For convergence with $N_{max}$, we see in fig.(12), the convergence is not as great as $\ell$. But since the general area of peak variation and best fit variation remains close to experiment, we can conclude that the peaks do exist and are not numerical artefacts.

The averaged minimum values of section 5 also vary with $N_{max}$ and $L_{max}$ as given in fig 13. We choose to calculate the minimum values for $N_{max}$ = 14 and $L_{max}$ = 19 as the values obtained and the S-matrix behaviour(regge, $\mathcal{E}$ and $\overline{S_R(\rho_1\|\rho_2)}$) are very similar to $N_{max}$ = 16 and $L_{max}$ = 25. Position of the minimum (as given in fig 4) barely changes with both $N_{max}$ and $L_{max}$.

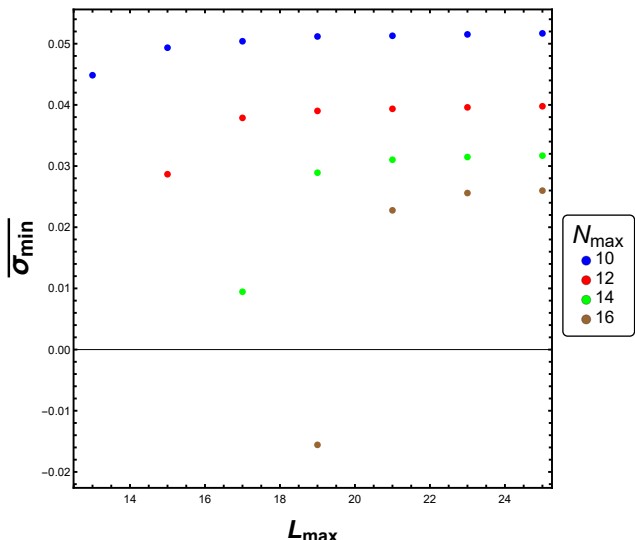

Figure 13: Variation of Average minimimum for $s_0$ = 0.3 and $s_2$ = 1.5 with $N_{max}$ and $L_{max}$

# E    Details for $\mathcal{E}$ and $\bar{\sigma}$

## E.1    Entanglement Power

We shall briefly describe the derivation of Entanglement Power for S-matrices following [15], [14] and [8]. Since we shall be dealing with isospin indices only, we can write,

$$|k\hat{n}, a_1; -k\hat{n}, a_2\rangle \equiv |a_1\rangle \otimes |a_2\rangle, \tag{E.1}$$

where $\hat{n}$ is a unit 3-vector and $a_1, a_2$ are isospin indices. The initial state is defined as,

$$|\psi_i\rangle := \hat{R}(\Omega_1) \otimes \hat{R}(\Omega_2) |p\hat{z}, a_1; -p\hat{z}, a_2\rangle \tag{E.2}$$

where $\hat{R}(\Omega_i)$ is rotation in the isospin space of the $i^{th}$ particle. In terms of usual spherical polar coordinates $(\theta_i, \phi_i)$, the rotation operator $\hat{R}(\Omega_i)$ is [8]

$$\hat{R}(\Omega_i) = \begin{pmatrix} e^{i\phi_i}\cos^2\left(\frac{\theta_i}{2}\right) & -\frac{e^{i\phi_i}\sin(\theta_i)}{\sqrt{2}} & e^{i\phi_i}\sin^2\left(\frac{\theta_i}{2}\right) \\ \frac{\sin(\theta_i)}{\sqrt{2}} & \cos(\theta_i) & -\frac{\sin(\theta_i)}{\sqrt{2}} \\ e^{-i\phi_i}\sin^2\left(\frac{\theta_i}{2}\right) & \frac{e^{-i\phi_i}\sin(\theta_i)}{\sqrt{2}} & e^{-i\phi_i}\cos^2\left(\frac{\theta_i}{2}\right) \end{pmatrix}. \tag{E.3}$$

The final state is defined using the S-matrix as,

$$
\begin{aligned}
|\psi_f\rangle &:= \frac{1}{w(p)^2} \sum_{\substack{c_1,c_2 \\ b_1,b_2}} |p\hat{n}, c_1; -p\hat{n}, c_2\rangle \langle p\hat{n}, c_1; -p\hat{n}, c_2 | S | p\hat{z}, b_1; -p\hat{z}, b_2\rangle \langle p\hat{z}, b_1; -p\hat{z}, b_2 | \psi_i\rangle \\
&= \frac{1}{w(p)} (2\pi)^4 \delta^{(4)}(0) \sum_{\substack{c_1,c_2 \\ b_1,b_2}} |p\hat{n}, c_1; -p\hat{n}, c_2\rangle \mathcal{S}^{c_1 c_2}_{b_1 b_2}(s, \cos\theta) \langle b_1 | \hat{R}(\Omega_1) | a_1\rangle \langle b_2 | \hat{R}(\Omega_2) | a_2\rangle \quad \text{(E.4)}
\end{aligned}
$$

where we have used the notation,

$$
\langle p\hat{n}, c_1; -p\hat{n}, c_2 | S | p\hat{z}, b_1; -p\hat{z}, b_2\rangle = (2\pi)^4 \delta^{(4)}(0) \, \mathcal{S}^{c_1 c_2}_{b_1 b_2}(s, \cos\theta), \tag{E.5}
$$

and the inner product,

$$
\langle k\hat{n}, b_1; -k\hat{n}, b_2 | \psi_i\rangle = w(k) \langle b_1 | \hat{R}(\Omega_1) | a_1\rangle \langle b_2 | \hat{R}(\Omega_2) | a_2\rangle, \tag{E.6}
$$

Using this final state, we can define the total density matrix $\rho_{\psi_f} = \bar{\mathcal{N}} |\psi_f\rangle \langle \psi_f|$. Evaluating in the isospin basis, we get,

$$
\left(\rho_{\psi_f}\right)^{b_1 b_2}_{c_1 c_2}(s, \cos\theta) = \frac{\sum_{x_1,x_2} \sum_{y_1,y_2} \mathcal{M}^{b_1 b_2}_{x_1 x_2}(s, \cos\theta) \left[\mathcal{M}^{y_1 y_2}_{c_1 c_2}(s, \cos\theta)\right]^* \hat{R}(1)^{b_1}_{a_1} \hat{R}(2)^{b_2}_{a_2} (\hat{R}(1)^{c_1}_{a_1} \hat{R}(2)^{c_2}_{a_2})^*}{\sum_{z_1 z_2} \sum_{x_1,x_2} \sum_{y_1,y_2} \mathcal{M}^{z_1 z_2}_{x_1 x_2}(s, \cos\theta) \left[\mathcal{M}^{y_1 y_2}_{z_1 z_2}(s, \cos\theta)\right]^* \hat{R}(1)^{z_1}_{a_1} \hat{R}(2)^{z_2}_{a_2} (\hat{R}(1)^{y_1}_{a_1} \hat{R}(2)^{y_2}_{a_2})^*}
\tag{E.7}
$$

where, $\hat{R}(1)^a_b = \langle a | \hat{R}(\Omega_1) | b\rangle$ and we assume that the scattering is strictly in non-forward direction. The reduced density matrix is defined using, $\bar{\rho}_1 = \text{tr}_2 \, \rho_{\psi_f}$, which is used in eq.(4.2) in the main text, and where $d\Omega_i = \sin\theta_i d\theta_i d\phi_i$. The general expression in terms of the amplitudes is quite hideous but near threshold, we find the following somewhat simpler expression for $\mathcal{E}$:

$$
\mathcal{E} = 1 - \frac{1}{16\pi^2} \int_0^{2\pi} d\phi_1 \int_0^{2\pi} d\phi_2 \frac{E_N(\phi_1, \phi_2)}{E_D(\phi_1, \phi_2)}, \tag{E.8}
$$

where with $\chi_1 = \cos(\phi_1 - \phi_2), \chi_2 = \cos(\phi_1 + \phi_2)$

$$
\begin{aligned}
E_N(\phi_1, \phi_2) &= 54(a_0^{(2)})^4 \left(1 + 6\chi_1^2 + \chi_1^4\right) + 48 \, (a_0^{(2)})^2 (a_0^{(0)} - a_0^{(2)}) (a_0^{(0)} + 2a_0^{(2)}) (3\chi_1^2 + 1)\chi_2^2 \\
&\quad + 16 \, (a_0^{(0)} - a_0^{(2)})^2 \left[(a_0^{(0)})^2 + 2a_0^{(0)} a_0^{(2)} + 3(a_0^{(2)})^2\right]\chi_2^4
\end{aligned}
\tag{E.9}
$$

and

$$
E_D(\phi_1, \phi_2) = 3[3(a_0^{(2)})^2(1 + \chi_1^2) + 2(a_0^{(0)} - a_2^{(2)})(a_0^{(0)} + a_2^{(2)})\chi_2^2]^2. \tag{E.10}
$$

The $\phi_1, \phi_2$ integrals cannot be carried out analytically even for this case. Nevertheless, using the NMaximize and NMinimize commands in Mathematica, one can with some effort show that

$$
0.14 \lesssim \mathcal{E}_{s \approx 4} \lesssim 0.67. \tag{E.11}
$$

The upper limit can be analytically derived to be $2/3$. In general, for $d$-dimensional "spin"-space the upper bound [15] for our distribution defined through eq.(4.2) [8] is $1 - 1/d$ and our result follows

---

[8]In [15], a stronger upper bound of $1/2$ exists for uniform distribution. Our averaging is not using a uniform distribution and our upper bound simply follows from the known [15] upper bound on the linear entropy $1 - \text{tr}_1\rho^2 \leq 1 - 1/d$.

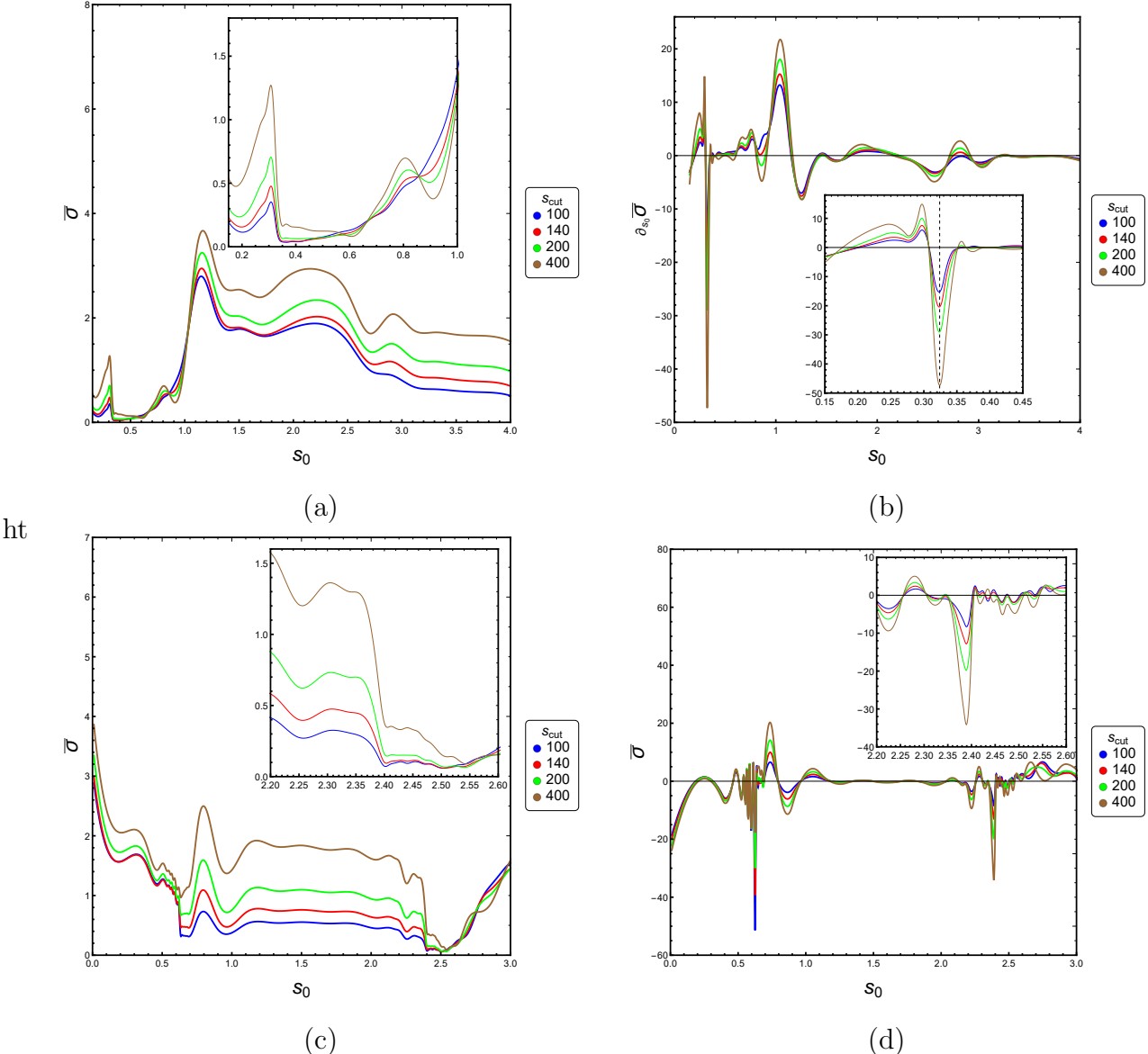

Figure 14: (a) Variation of $\bar{\sigma}^{\pi^0\pi^0}$ with $s_0$ on the upper boundary and (b) Variation of $\partial_{s_0}\bar{\sigma}$ with $s_0$ on the upper boundary for different $s_{cut}$. (c) Variation of $\bar{\sigma}$ with $s_0$ on the lower boundary and (d) Variation of $\partial_{s_0}\bar{\sigma}$ with $s_0$ on the upper boundary for different $s_{cut}$.

since $d = 3$. This is a non-trivial check on our calculations since it is unobvious from the complicated form of eq.(E.8) how this arises. In our numerical exploration we have not found any violation to the upper limit although we do not have a direct proof for any $s$.

## E.2 Drop in $\bar{\sigma}$

In fig.(14), we show the behaviour of $\bar{\sigma}^{\pi^0\pi^0}$ as a function of $s_0$ for different choices of $s_{cut}$. While the actual location of the global minimum near **A** appears to shift to the right, the sharpest drop occurs at the same point near **A** as is clear from the plot of $\partial_{s_0}\bar{\sigma}$. The situation is similar near **C**.

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
