# Peer review of "Selection rules for the S-Matrix bootstrap"

_SciPost Physics_

## Round 2 · Referee Report · Anonymous (Referee 1) · 2021-4-9

Strengths

1-The authors have studied the space of scattering amplitudes in 3+1 dimensions compatible with crossing, unitarity and analyticity and low energy QCD, focusing in particular on the allowed region of Adler zeros.

2-The authors have uncovered a number of interesting connections between geometrical features of the space of Adler zeros and physical observables such as cross sections, entanglement power and Regge behaviour.

Report

In the manuscript "Selection rules for the S-matrix bootstrap" the authors have explored two observables, the integrated cross section and the entanglement power, to single out physical theories out of the allowed space of all possible amplitudes. The paper is well written and the numerical results are intriguing and encouraging. Therefore, I recommend it for publication in SciPost if the following comments and questions are addressed.

Requested changes

1-The authors give great importance to the Lovelace-Shapiro model, but some clarifications might help appreciate it more. For instance: LS passes close to the points where Regge linearity shows up, but only A, B, C, not D; on the other hand at B the authors find that even and odd trajectories have different slopes but LS not. Is there an intuition behind these inconsistencies?
Moreover, if one uses fpi, and the values of the isospin 0 and 2 scattering lenghts, does that completely fix LS parameters? If so, is it an interesting point?
What is the meaning of the non-unitarity of the LS model mentioned in sec 2?

2-In fig.1, do the 1 and 2-loop Adler zeros have error bars? I would expect that they depend on the low ChiPT low energy constants which are known with some errors.

3-In sec. 3 the authors claim they do not have elastic unitarity, in that case why a peak in the amplitude should correspond to a resonance? Do they observe at least approximate elastic unitarity?

4-The sigma resonance is not expected to have a Breit Wigner form, how reliable is the position determined by fitting the partial amplitudes with the position of the zero in complex plane? How much do the slopes of the linear trajectories around A, ..., D depend at all on the position of sigma?

5-When the authors say standard model, they mean standard model with m_up=m_down?

6-Is there is a typo in the label in fig. 14d?

7-In fig. 2b the authors show slopes for even and odd trajectories, which colour is odd and which one is even?

---

## Round 2 · Referee Report · Yifei He (Referee 2) · 2021-4-11

Strengths

1, well-structured and clearly written 2, helpful appendices on supplementary materials 3, the idea of examining appearance of Regge trajectories is new and the result of a selection rule for such behavior is interesting

Weaknesses

1, some figures are not clearly described 2, the convergence of the numerical results is not very satisfactory

Report

In the paper the authors follow their previous work [8] of examining the space of pion scattering as characterized by the Adler zeros in the S-wave of isospin 0, 2 amplitudes. This was carried out using the S-matrix bootstrap setup of reference [6] where the mass of rho resonance is imposed. In addition, they imposed chiral perturbation theory motivated inequalities of the S- and D-wave scattering lengths which in their previous work [8] have been given a quantum information theoretic interpretation. The result is a "river" -shaped allowed region in the two dimentional Adler zero space.

The novelty of this paper is to make connections with Regge theory. In particular, they found regions on the boundary of the allowed space displaying Regge trajectories. The resonance masses are identifi?ed by fitting the peaks of the partial waves. An interesting feature is that the linear Regge trajectories happens at the kink-like places where the Lovelace-Shapiro model intersect the space. This further motivates the author to search for a selection rule for Regge behavior and they found that along the boundary of the space, the special points displaying linear Regge trajectories correspond to minima of an averaged total cross section and a quantity called entanglement power. A further exploration is done by imposing such selection rules in the bootstrap program which probes inside the space and this results in linear Regge behavior.

The idea of the paper is new and the result is very interesting. This should lead to new directions in the higher dimensional S-matrix bootstrap program and make closer contact with experiments and phenomenology.

I will recommend for publication after the following questions/comments are answered.

Requested changes

1, it would be good to include better references to the experimental results used in a few places in the paper (e.g. captions of figure 8, figure 2). The current references make it hard to the reader to consult the original data.

2, in section 6 Future directions the authors mentioned better numerical approaches and referred to [20] which extended the dual problem proposed in 1909.06495. Since [20] is cited, it would be appropriate to also cite 1909.06495 where the proposal was originally made.

3, figure 12 and its caption are confusing. In figure 12(a) the green bands shows big deviation in Lmax but in the text the author claims this shows a good convergence in angular momentum truncation which is strange. The caption also refers to black squares in (a) which are experimental data but there are no black squares. In addition, one of the experimental data point in (a) and (b) with J=4 is shifted for unknown reason. On the other hand, the green bands in (b) looks almost the same as (a) and the claim there is that this is the deviation in the Nmax truncation. The whole figure is very hard to understand.

Related to the last point above, I also have some technical question/comment regarding the numerics: The authors justified the method of resonances detection by claiming that elastic unitarity is not true. Indeed elastic unitarity is not imposed as a constraint in the S-matrix bootstrap, but it is generically obeyed as a result of the optimization, especially for low partial waves at low energy. It is unclear from the paper whether elastic unitarity is satisfied here which -- if not -- would imply a bad convergence in the numerics.

typos: 1, third line of page 14: fig. 4 -> fig. 7 2, fig. 14 (d) the vertical axis should be $\partial_{s_0}\bar{\sigma}$

---

## Round 2 · Referee Report · Anonymous (Referee 3) · 2021-4-16

Strengths

1- Detailed numerics 2- Focus on emergence of Regge linearity 3- Set of distinct intriguing coinciding observations

Weaknesses

1- Lack of clarity in the presentation of the text and the figures 2- Minor missing point on absence of linearity away from special locus (see point 1. below)

Report

Dear Editor,

In this text, the authors provide a numerical study of the pion-pion S-matrix bootstrap, building up on results by the same authors and collaborators in 2006.12213 (ref. [8]). They explore the set of consistent S-matrices, and search in particular for the emergence of linear Regge trajectories. This leads them to postulate "selection rules" for the S-matrices (section 4).

One of the main strengths of the paper is this focus on the emergence of linear Regge trajectory, which is one of the important questions of this program. It is remarkable that some amount of linearity can be achieved within this framework and, moreover, that it seems to coincide with "kink-type" features of the construction. I find the combination of the various, numerically robust, evidence coming several directions, makes the paper worth being published in Scipost.

The paper is written in a concise format, with main results quoted within the first 10 pages, followed by explanatory appendices. However, the overall clarity of the text could, in some places, be improved. Below are a few suggestions to this end.

Most of my related remarks below are rather minor, but I have two more important concerns, regarding: the evidence for the emergence of Regge linearity, and the meaning of the "new river" in figure 11.a, appendix D.

  1. I have a presentation worry about the emergence of linearity. To back the claim that linearity only emerges in the green regions of fig. 1, I only find evidence in fig. 10, far in the appendix, while it seems a very important piece of evidence. Could it be possible to at least refer to this graph sooner in the text ? Relatedly, why does fig. 5 only show even spins, when fig. 10 shows both even and odd. Is it possible to produce a plot which spans the whole boundary ? If not (because the numerics would be too hard), please explain why and why you believe that there is no linearity away from ABCD.

  2. I may have misunderstood the meaning of the new river (dashed grey) in figure 11.a. It seems to wash away the "kink" nature of region A, and, more worryingly, to push out of the allowed region the intersection of the minimum curve of figure 4.b with the boundary (and the LS amplitude). Could the authors argue why one should not worry about this ? It might decrease the strength of the coincidence, exemplified again by the near intersection of the three curves in figure 4.a.

Before moving to a detailed list of suggestions, I also had the following general observation. The drop in entanglement entropy seems not particularly motivated, and, as the authors say, could well be attributed to the corresponding decrease of cross-section, indicating smaller interactions and hence smaller entanglement entropy. While I understand that this could be the subject of another study, it might also be that I have missed a basic point. In that case, it would be nice if the authors could explain more clearly why they think that their rules 1. and 2. are not a consequence of each other. I would leave it to the authors to comment on this or not.

Requested changes

Here are a list of problems to be fixed or requested further modifications: (for 1. and 2., see above)

  1. there is a latex problem with hyperref package, the footnotes point to the first page of the pdf,
  2. phrasing of the intro could be improved, a lot of "quite" and "decent" conveys a feeling of vagueness, inappropriate for a numerical work, (also appear elsewhere in the text, searching for "quite" and improving precision on these words would be nice)
  3. In general, could the authors argue why they think that it is nice at all to have intersection with the LS amplitude, which is not unitary here ?
  4. Relatedly, for readability purposes, it might be worth recalling the fact that the (non-unitary) LS amplitude crosses the allowed (unitary) region is not a contradiction, because "allowed region" simply means that unitary S-matrices can exist there (for instance this can be added within footnote 2).
  5. The overall exposition of the main results in the introduction could be improved; for instance it might be stressed with explicit sentences "the main results of these papers are...", or "we find that...": this would help the reader identifying more precisely what the authors think is important,
  6. Quantities in eq. 2.1 are not defined
  7. it is slightly unclear to me what footnote 3 means specifically here, it sounds more like a general comment on cut-constructibility and dim. reg., if I understood correctly ? Otherwise, the authors might want to give a bit more details here, as there is no space constraint.
  8. fig. 1 : on a printed version, the "green" region is almost invisible. I'm not sure what can be done about this, however. To the least, adding in the caption "The green regions on the boundary, near A and C, .."
  9. Section 3., in the description of the results for D., it is not clear what is the meaning and implication of the fact that "l=8 is missing".
  10. Last paragraph of page 5, same problem of precision of language with "somewhat" being used, please make more precise claims or remove those adverbs altogether (I am aware that the appendices provide details but this just sounds vague)
  11. fig. 2b: legend or definition for black/red curves is missing
  12. eq 4.1: at this point, the authors have not yet defined their procedure to find the s_2 that minimizes \sigma, hence, it seems that \sigma there should depend on both s_0 and s_2 ?
  13. Related to my question above, and the comment in rule 2. on page 8, what exactly is the reader supposed to compare to fig. 3 to see that the relation between sigma and E is more subtle ? fig 14 ?
  14. relatedly, why is fig 14 not in main text ? or maybe a variation of it, the analogue of fig. 3 but for sigma ?
  15. could the authors justify the choice of s_cut = 375 ?
  16. S_R and P_g in 5.1 are not defined
  17. could the authors comment somewhere in the text on whether or not they find the results of fig. 8 good or not and why ?
  18. fig. 9 is very unclear, what are the dots, why is the value of s_0 on top of those graphs without mention of s_0, shouldn't the dots have two colors for even and odd ?
  19. fig. 10: it is unclear to me why the points are joined
  20. fig 12: what do the different values of Lmax and Nmax correspond to, in the graph ?
  21. naive question: why do the authors stop at low values of spin, in fig. 8 for instance ? are the other not captured because of the truncation of Lmax ?

---

## Editorial Decision

resubmitted